# Clinical and Functional Outcomes in Faller and Non-Faller Older Adults Clustered by Self-Organizing Maps: A Machine-Learning Approach

**Milena L. S. Almeida** [1], **Aline O. Cavalcanti** [2], **Rebeca Sarai** [3], **Mateus A. Silva** [3], **Paulo R. V. Melo** [3], **Amanda A. M. Silva** [2], **Rafael R. Caldas** [3], **Fernando Buarque** [3] **and Francis Trombini-Souza** [1,2,*]

1 Department of Physical Therapy, University of Pernambuco, Petrolina 56328-900, PE, Brazil; milena.lopesa@upe.br
2 Master's and Doctoral Programs in Rehabilitation and Functional Performance, University of Pernambuco, Petrolina 56328-900, PE, Brazil; aline.oliveirac@upe.br (A.O.C.); amanda.silva@upe.br (A.A.M.S.)
3 Polytechnic School of Engineering, University of Pernambuco, Recife 50720-001, PE, Brazil; rsas@ecomp.poli.br (R.S.); mas3@ecomp.poli.br (M.A.S.); paulovarjal@gmail.com (P.R.V.M.); rrc@ecomp.poli.br (R.R.C.); fbln@ecomp.poli.br (F.B.)
* Correspondence: francis.trombini@upe.br; Tel.: +55-87-3866-6470

**Featured Application: Increased cognitive demands during motor tasks emerged as a critical discriminator of functional patterns among older adults, shedding light on the intricate interplay between cognitive function and motor performance in fall prevention strategies in community-dwelling older adults.**

**Abstract:** A wide range of outcomes makes identifying clinical and functional features distinguishing older persons who fall from non-fallers challenging, especially for professionals with less clinical experience. Thus, this study aimed to map a high-dimensional and complex clinical and functional dataset and determine which outcomes better discriminate older adults with and without self-reported falls. For this, clinical, functional, and cognitive outcomes of 60 community-dwelling older adults classified as fallers and non-fallers were selected based on self-report of a single fall in the last 12 months. An unsupervised intelligent algorithm (Self-Organizing Maps—SOM) was used to cluster and topographically represent the data studied. The SOM model mapped and identified two different groups (topographic error: 0.00; sensitivity: 0.77; precision: 0.42; accuracy: 0.53; F1-score: 0.55) based on self-report of a single fall. We concluded that although two distinct groups were mapped and clustered by the SOM, participants were not necessarily fallers or non-fallers. The increased cost of cognitive demands regarding a motor task (Timed Up and Go Test) and the effect of the Trail Making Test (TMT) Part B regarding TMT Part A could discriminate the functional and cognitive patterns in community-dwelling older adults. Therefore, in clinical practice, identifying patterns involving the interaction between cognition and motor skills, even in once-only faller older adults, can be an efficient approach to assessment and, consequently, to compound intervention programs to prevent falls in this population.

**Keywords:** falls; artificial intelligence; functional mobility; unsupervised algorithm; older adults

## 1. Introduction

Functional mobility impairment is a crucial factor associated with falls in older adults [1]. Physical injuries and psychological impacts resulting from falls can lead to a loss of autonomy, disability, low quality of life, and even increased incidence of death in older adults [2]. Those with recurrent self-reported falls exhibit reduced muscle strength, impaired body balance [1], and a decline in cognitive function, including executive function, attention, memory, visuospatial capacity, and processing speed [2].

Several studies have tried to discover one or more clinical and functional performance outcomes to identify older adults at greater risk of falling [3,4]. However, given the large data volume in geriatric assessment, selecting outcomes that better represent cognitive, executive, intellectual, and motor functions is challenging in clinical practice [5]. An umbrella review of instruments assessing gait, balance, and functional mobility concluded that no univariate outcome could predict the risk of falls in older adults since this event has multifactorial causes [6].

Furthermore, identifying nuances from many outcomes can be challenging, especially for professionals with less clinical experience [7]. Thus, this task becomes relevant not only to understanding the causes related to the risk of falls [6] but also to analyzing the applicability of intelligent and multivariate methods in the assessment and grouping of individuals with similar characteristics [8], i.e., clinical and functional patterns related to falls in older adults.

Machine learning (ML) is at the core of the Big Data Revolution because of its ability to learn from data and provide data-driven insights, decisions, and predictions. It promises to transform how we live, work, and think by optimizing processes, empowering insight discovery, and improving decision-making [9]. In this context, several ML-based models, such as random forest (RF), logistic regression (LR), support vector machine (SVM), light gradient boosting machine (LightGBM), and extra trees (ET) have been applied to analyze functional dependency in basic activities of daily living (ADL) and instrumental activities of daily living (IADL) in middle-aged and older adults [10]. ML-based models have also been used to analyze motion, clinical, cognitive, and functional outcomes [11,12]. Ensemble techniques, such as AdaBoost, clustering models (e.g., k-means), and SVM, have presented promising measurement properties in identifying and differentiating the functional features in community-dwelling older adults undergoing exercise protocols [12].

From a wide range of ML methods, the Self-Organizing Map (SOM) [13] is an unsupervised artificial neural network capable of reducing data with great dimensionality and complexity [14,15]. The SOM can also classify individuals more efficiently regarding computational resource time spent than traditional statistical methods. Furthermore, as it is an unsupervised method, it does not require specific and advanced training for use in the clinical context [8].

Thus, the present study aimed to: (i) map a high-dimensional and complex clinical and functional dataset; and (ii) determine which outcomes better discriminate community-dwelling older adults with and without self-reported falls in the previous 12 months. The premises of this study are that: (i) self-organizing mapping will accurately identify the participants with self-reported falls based on clinical and functional outcomes; and (ii) the costs between physical and cognitive performance will be more relevant among the other outcomes addressed in this study to identify and differentiate community-dwelling older adults with and without self-reported falls.

## 2. Materials and Methods

### 2.1. Study Design and Ethical Aspects

This observational, analytical, cross-sectional study was designed and developed following the guidelines for Strengthening the Reporting of Observational Studies in Epidemiology (STROBE) [16].

The data analyzed in this study came from the baseline of the first version of the Study on Falls in Older Adults (EQUIDOSO-I) [17]. This study was developed following the Declaration of Helsinki, the recommendations of the World Health Organization, the General Data Protection Law, and the International Committee of Medical Journal Editors. This study was approved by the Research Ethics Committee of the University of Pernambuco (CAAE: 71192017.0.0000.5207, Opinion Number: 2.415.658). All participants gave their written agreement to participate in this study.

### 2.2. Participants

A total of 60 community-dwelling older adults aged between 60 and 80 participated in this study. Of these 60 participants, 22 who self-reported falling in the last 12 months were labeled "fallers", and 38 were considered "non-fallers". This definition considered that fallers are usually distinguished from non-fallers in clinical and scientific practice by their self-reporting of falls [18] in the last 12 months [19,20].

Participants with a score $\geq$ 52 points (up to a maximum of 56 points) on the Berg Balance Scale [21], scoring $\geq$ 24 points on the Mini-Mental State Exam (MMSE) (score maximum of 30 points) for those with >4 years of formal education or $\geq$18 points those with <4 years of formal education [22], and who were able to walk uninterruptedly for a distance of 60 m at a self-selected speed of at least 1 m/s (without the help of third parties or walking aids) were included. Potential participants were excluded if they: (i) had any restrictions in postural balance or cognition; (ii) self-reported two or more falls in the last 12 months; (iii) were participating or had participated in any regular and structured exercise program physical exercise two or more times a week in the last six months; (iv) had any chronic health condition for which physical exercise was contraindicated; (v) had any upper or lower limb fracture in the last six months; (vi) had evidence of any surgical procedure on the knees, ankles, or hips or muscle injuries in the last six months; or (vii) had a diagnosis of uncontrolled diabetes.

For the sample size calculation in the EQUIDOSO-I study [17], the gait speed under dual-task [21] (the primary outcome) was adopted. Aiming to achieve a minimum clinically important difference of 0.05 m/s, an effect size of 0.20 [23], a test power of 95% $(1 - \beta)$, an alpha of 0.05, and a repeated-measures F-statistic design, with between- and within-subject and interaction effects were considered. The initial sample size of EQUIDOSO-I was 48 individuals, which was increased by 20% to account for potential sample loss. Thus, a total of 60 participants was assessed to adequately provide the required power for traditional statistical tests employed in the EQUIDOSO-I study. The sample size was calculated using the G* Power 3 program [24].

### 2.3. Database

The data analyzed in this study were the walking speed under single gait and dual-tasking [25], the conventional [26] and cognitive Timed Up and Go test (TUG) [27], the TUG effect, the time obtained in the Five-Time Sit-to-Stand Test [28], the distance from the Anterior Functional Reach Test [29], the time during the Trail Making Test (TMT) A and B [30], TMT B cost, Black and White and Color Stroop Test [31], the Colored Stroop Test effect, the scores from the Brazilian versions of the Activities-Specific Balance Confidence Scale (ABC Scale) [32], Falls Efficacy Scale–International (FES-I) [33], and the Geriatric Depression Scale [34].

Thus, in this secondary analysis, using the SOM, we employed a high-dimensional and representative dataset of 60 community-dwelling older adults aged 60 to 80, with or without a fall history, to guarantee the quality of the final clustering model. This methodological feature was sufficient to ensure that the SOM attained sufficient convergence, improved data-innate patterns, improved participant grouping, and stability and interpretability of the generated clusters.

Clinical and functional outcomes considered most relevant for this study were selected based on previous analyses carried out by two physiotherapists (F.T.S. and R.R.C.) with more than ten years of experience in functional analysis of older adults. Variables with autocorrelation values equal to or greater than 0.80 among themselves and outcomes with lower weights were excluded.

### 2.4. Analysis

The SOM algorithm was programmed using the open-source application Jupyter Notebook, the Python programming language (version 3.8) and the os, numpy, pandas, sompy, matplotlib.pyplot, BmuHitsView, HitMapView, sklearn.metrics, classification_report li-

braries, and View2D for file manipulation, data mining, SOM calculation, specific statistical metrics and graphical plotting, respectively. These procedures provided a set of topological maps whose visual representation demonstrates the spatial arrangement of the processing units distributed along two-dimensional matrices with a spatial resolution of $5 \times 8$ neurons (height $\times$ width), totaling 40 neurons.

Models were developed using the SOM algorithm and combinations of chosen outcomes to select the one that best preserved the implemented input data and obtained valid values in evaluating the algorithm's performance concerning the generated grouping.

First, two metrics within the SOM algorithm quantify the degree of similarity between the original data set and the obtained clustering. Topographic error quantifies the discrepancy (or preservation of similarity) between the spatial organization of neurons (processing units) and the topological relationships in the original data. A smaller model's value indicates that the spatial organization of neurons in the topological map adequately preserves the existing structure in the input (original) data [35].

The quantization error measures the distance between each data sample and its winning neuron (BMU—Best Matching Unit) in the SOM; then, the average of these distances is calculated for all samples. It is a measurement that denotes the map's resolution. A smaller quantization error indicates that the SOM neurons are being more precisely tuned to represent the patterns in the original data [35].

The Adjusted Rand Index (ARI) was used to evaluate the quality of the clusters generated by the SOM. Values close to 0 indicate that the clusters are independent, and negative values suggest a disagreement between the clusters [36,37]. The degree of similarity among the assigned groupings was graded by Adjusted Mutual Information (AMI), which varies from 0 to 1, in which 1 indicates a perfect match and 0 indicates a random match [36].

Accuracy represents the SOM's ability to correctly group data into clusters that correspond to the actual class of the data (true data labels) or the negative class.

The F1-score provides a way to balance accuracy and sensitivity by considering both false positives and negatives, which is useful when a significant imbalance exists between classes. High values indicate a good balance between precision and recall. In contrast, a low F1-score suggests that there are problems with data clustering. Precision, recall, and F1-score can provide a more complete view of the quality of the classification [37].

Descriptive statistical analyses and univariate inferences of the demographic, anthropometric, and functional variables of the falling and non-falling older adult groups were performed using the *t*-test (variables with parametric distribution) and the Mann–Whitney U test (non-parametric variables). The Kruskal–Wallis test for independent samples was used to initially compare the four age groups. All statistical analyses were performed in the Statistical Package for Social Sciences (IBM SPSS, version 22; IBM Corp, Armok, NY, USA), adopting a significance level of 5%.

## 3. Results

Firstly, participants were divided into four age groups to verify any significant difference in physical and cognitive function inherent to age, as follows: 60–64 years old (y.o.) ($n = 19$; 31.7%), 65–69 ($n = 21$; 35.0%), 70–74 y.o. ($n = 14$; 23.3%), and 75–80 y.o. ($n = 6$; 10.0%). The Kruskal–Wallis test for independent samples confirmed no significant functional and cognitive differences as follows: gait speed ($p = 0.446$), conventional TUG ($p = 0.455$), cognitive TUG ($p = 0.621$), dual-task effect (TUG) ($p = 0.398$), Sit-to-Stand from a chair 5 Times ($p = 0.245$), and Anterior Functional Reach Test ($p = 0.282$).

Table 1 shows no significant difference between faller and non-faller participants regarding demographic, anthropometric, functional, and clinical outcomes.

The variables considered for each group (fallers and non-fallers) in the defined model are organized in the radar graphs shown in Figure 1. The length of each ray in this graphical representation is proportional to the magnitude of the variable for the data point. This figure shows that the data from the non-faller (label 0) and faller (label 1) groups presented

very similar graphical representations, reiterating the conventional statistical analysis that did not identify significant between-group differences, as shown in Table 1.

**Table 1.** Univariate comparisons of demographic, anthropometric, functional, and clinical outcomes of faller and non-faller groups.

| Variables | Fallers $n = 22$ | Non-fallers $n = 38$ | | |
|---|---|---|---|---|
| | n (%) | n (%) | | |
| Female | 19 (86.4%) | 33 (86.8%) | | |
| Male | 3 (13.6%) | 5 (13.2%) | | |
| | Mean (SD) or Median (Q1–Q3) | Mean (SD) or Median (Q1–Q3) | Cohen's *d* | *p*-Value |
| Age (years) | 65.50 (62.00–71.00) | 67.50 (64.00–70.25) | 0.70 | 0.210 § |
| Boby mass (kg) | 68.75 (63.37–74.55) | 59.17 (68.67–75.55) | 1.49 | 0.645 § |
| Height (m) | 1.54 (0.06) | 1.56 (0.07) | 0.31 | 0.187 * |
| BMI (kg/m²) | 29.55 (3.96) | 28.07 (5.03) | 0.32 | 0.244 * |
| Gait speed (m/s) | 1.32 (0.14) | 1.34 (0.16) | 0.13 | 0.512 * |
| Conventional TUG (s) | 9.15 (9.99–11.25) | 8.45 (9.55–10.90) | 0.72 | 0.398 § |
| Cognitive TUG (s) | 10.77 (12.20–14.12) | 10.47 (12.25–14.50) | 0.19 | 0.860 § |
| Effect of TUGCog (%) | −0.17 (−0.39−−0.08) | −0.40 (−0.25−−0.08) | 1.35 | 0.623 § |
| STS-5X (s) | 13.70 (11.55–15.72) | 14.35 (11.47–16.25) | −0.19 | 0.724 § |
| AFRT (cm) | 16.44 (3.35) | 16.73 (4.24) | 0.07 | 0.785 * |
| TMT-A (s) | 57.80 (44.87–66.47) | 63.69 (50.30–87.80) | −0.24 | 0.129 § |
| TMT-B (s) | 137.00 (92.67–224.25) | 176.71 (120.77–227.62) | −0.46 | 0.211 § |
| Effect of TMTB (%) | −141.91 (−274.31−−78.10) | −160.72 (−257.78−−77.67) | 0.14 | 0.914 § |
| BW Stroop Test (s) | 33.30 (30.40–40.60) | 35.90 (31.82–39.65) | −0.40 | 0.500 § |
| Color Stroop Test (s) | 87.40 (70.85–113.95) | 78.00 (65.46–91.35) | 0.38 | 0.308 § |
| Color Stroop Effect (%) | −153.24 (−243.37−−99.04) | −138.41 (−154.82−−86.20) | −0.19 | 0.133 § |
| ABC Scale (0 to 100%) | 71.14 (15.58) | 70.45 (20.35) | 0.04 | 0.892 * |
| FES-I (16 to 64 points) | 29.00 (21.00–30.00) | 25.00 (21.00–32.25) | 0.51 | 0.817 § |
| GDS (0 to 15 points) | 3.50 (2.00–5.00) | 3.50 (2.00–6.25) | 0.00 | 0.551 § |

SD: Standard deviation; Q1: Quartile 1; Q3: Quartile 3; BMI: Body mass index; TUG: Timed Up and Go test; STS-5X: Sit-to-Stand Test 5 Times from a chair; AFRT: Anterior Functional Reach Test; TMT: Trail Making Test; BW: Black and White; ABC: Activities-Specific Balance Confidence, FES-I: Falls Efficacy Scale–International; GDS: Geriatric Depression Scale. § Intergroup comparison performed using the Mann–Whitney U test, with central tendency and dispersion measures represented by the median and interquartile range, respectively. * Intergroup comparison performed using the *t*-test, with central tendency and dispersion measures represented by the mean and standard deviation, respectively.

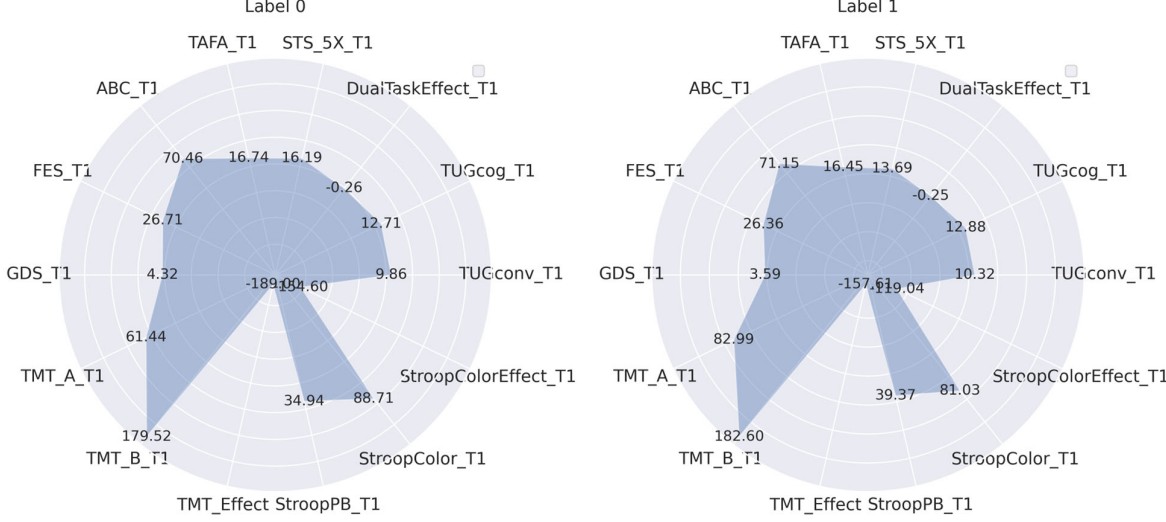

**Figure 1.** Clinical and functional profile of the faller (label 1) and non-faller (label 0) groups.

Figure 2 shows the result of the proposed model's confusion matrix. Of the 22 participants who self-reported falls in the last 12 months, the model classified 17 (77%) as fallers (sensitivity for class 1; fallers).

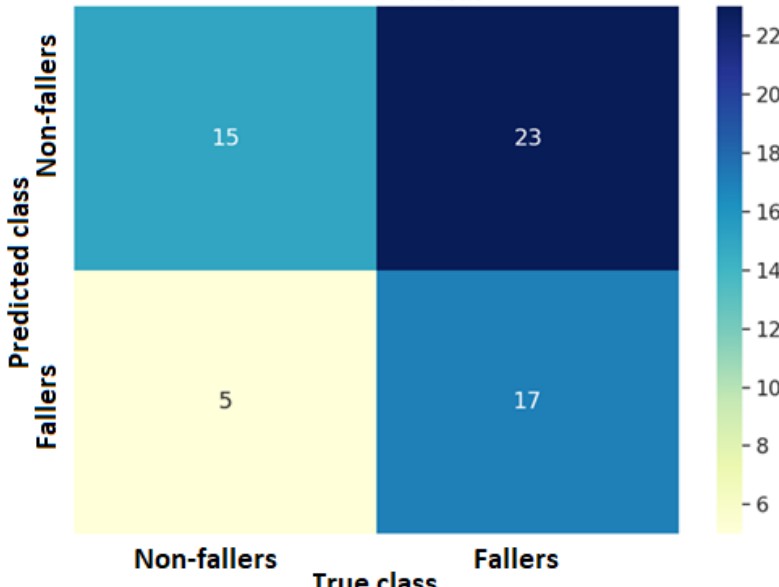

**Figure 2.** Confusion matrix regarding SOM algorithm clustering of faller and non-faller groups.

The results presented in Table 2 show that the model has better sensitivity in identifying fallers (0.77) and a lower sensitivity in identifying non-fallers (0.39). Regarding precision, the model showed better effectiveness in the proportion of non-falling positive examples (0.75). The F1-score is the harmonic mean of precision and sensitivity; this metric is helpful in classification problems with sample size imbalances as it considers both precision and recall, providing a more balanced view of model performance regardless of class imbalance. The model presented the highest F1-score for identifying fallers (0.55) compared to non-fallers (0.51). Accuracy is the proportion of cases correctly classified by the model concerning the total number of cases, for which the model presented 0.53. In addition, the model presented a low error and better preservation of the original data concerning the neurons represented by the topographic error (0.00). The ARI and AMI values showed a weak correspondence between clusters and true labels.

**Table 2.** Performance metrics for the SOM model.

| Parameter | Group | Metric Value |
|---|---|---|
| Topographic error | - | 0.00 |
| Quantization error | - | 2.62 |
| Adjusted Rand Index | - | −0.01 |
| Adjusted mutual information | - | 0.01 |
| Sensitivity (recall) | Fallers | 0.77 |
|  | Non-fallers | 0.39 |
| Precision | Fallers | 0.42 |
|  | Non-fallers | 0.75 |
| F1-score | Fallers | 0.55 |
|  | Non-fallers | 0.51 |
| Accuracy | - | 0.53 |

Figure 3 represents the distribution of neurons in each of the 14 functional variables individually, called conventional TUG (TUGconv_T1), cognitive TUG (TUGcog_T1), TUGCog effect (DualTaskEffect_T1), TSL-5X (STS_5X_T1), TAFA (TAFA_T1), ABC Scale (ABC_T1), FES-I (FES_T1) and EDG (GDS_T1), TMT-A (TMT_A_T1), TMT-B (TMT_B_T1), effect of

TMT-B (TMT_Effect), Stroop PB test (StroopPB_T1), Color Stroop test (StroopColor_T1) and Color Stroop effect (StroopColorEffect_T1). The result of this model showed that only the cognitive TUG, the effect of the cognitive TUG regarding the conventional TUG, and the effect of the TMT-B regarding the TMT-A were capable of signaling the existence of two groups in the analyzed data set. The more diametrically opposed the darker-colored neurons are, the greater the discriminative potential of a given outcome. In the topographic distribution concerning the cognitive TUG (TUGcog_T1), it is possible to see that the neurons representing the individuals with a pattern of the longer execution time of this dual-task test are concentrated in the lower left corner (dark yellow), while the participants who performed this test in a shorter time (better performance) are located in the upper right corner (dark blue), i.e., in opposite directions. In the topographic distribution of the effect of the cognitive TUG compared to the conventional TUG (DualTaskEffect_T1), the neurons representing the individuals with the highest cost pattern are concentrated in the lower left corner (dark blue), while those with the lowest cost are located in the upper right corner (green and yellow). Finally, the topographic distribution of neurons referring to the effect of the TMT-B regarding the TMT-A (TMT_effect) shows that individuals with less cognitive difficulty (dark yellow) are concentrated in the upper left corner, while participants with greater cognitive difficulty are located in the bottom right corner (dark blue).

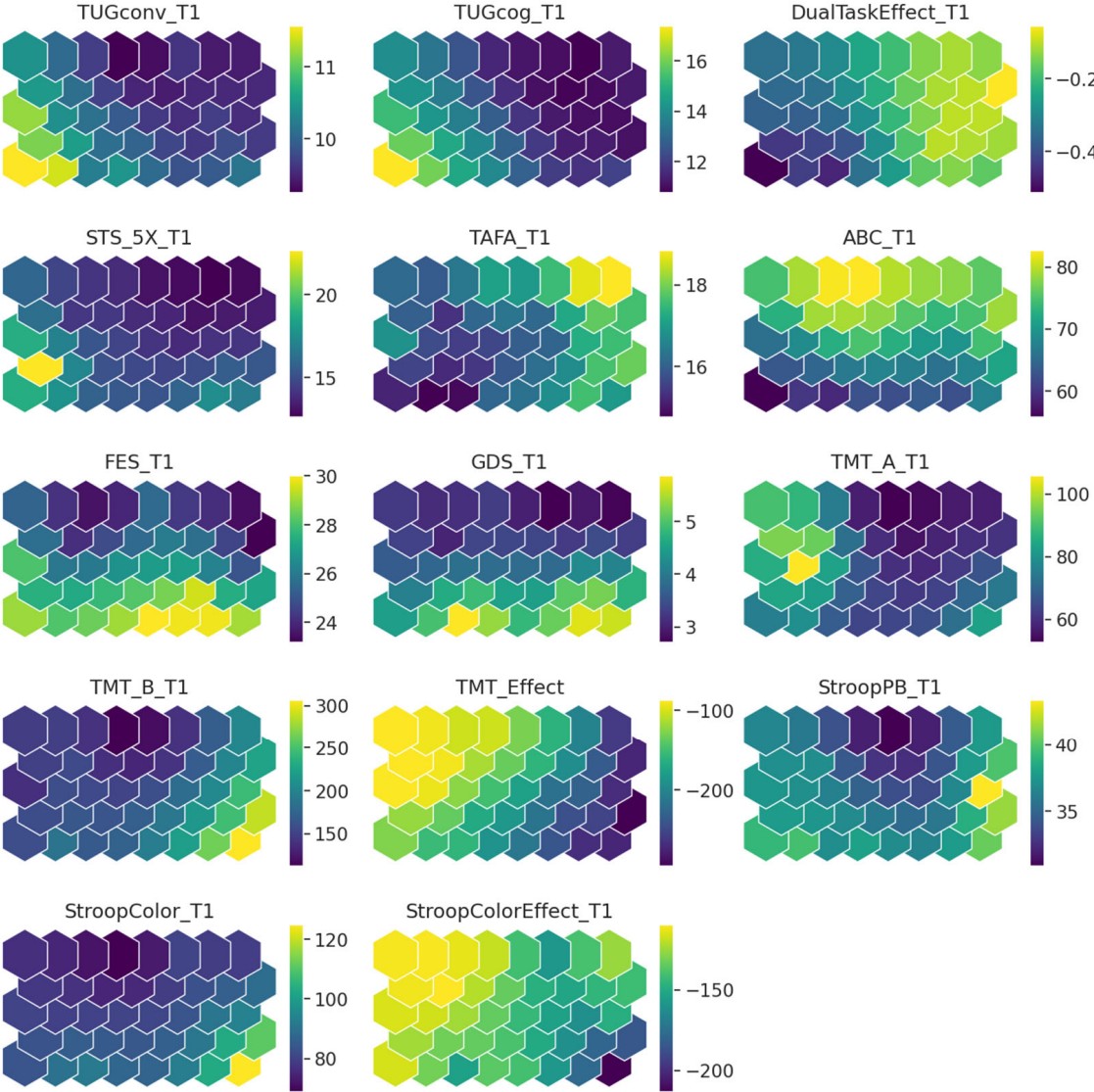

**Figure 3.** Topological maps showing the distribution of neurons for all functional outcomes.

## 4. Discussion

The first purpose of this study was to map a high-dimensional and complex clinical and functional dataset from community-dwelling older adults with and without self-reported falls in the previous 12 months. The premise concerning this aim was that the self-organizing mapping would accurately identify the participants with and without self-reported falls based on these data. However, although the main results of the model achieved a sensitivity of 77% in identifying participants with a self-report of a single fall, the accuracy was roughly 50%. This accuracy percentage demonstrates that the proportion of cases correctly classified by the model regarding the total number of cases was close to a random classification.

Although the clinical and functional data considered in this study allowed the SOM to make the topological distinction between the two groups, these results suggest that the participants were not necessarily "fallers" and "non-fallers" as previously labeled based on self-report of a single fall in the last 12 months. Hence, these results lead us to refute the first premise of this study.

A possible explanation for the accuracy of approximately 50% may have been the imbalance between the number of participants in the two groups, as the group of fallers consisted of 22 participants while the non-fallers group contained 38 older adults. Although this imbalance could negatively interfere with accuracy, this proportion is representative of the prevalence of fallers in the general population, especially in Brazil [38]. However, when analyzing the results of models with balanced samples (22 fallers and 22 non-fallers), no significant improvement in the magnitude of accuracy was observed. Furthermore, the F1-score value was not significantly modified with or without the imbalanced sample. It is worth highlighting that the F1-score (the harmonic mean of precision and sensitivity) is a valuable metric in classification problems with sample size imbalances as it considers both precision and recall, providing a more balanced view of model performance regardless of class imbalance. Therefore, the model with a sample imbalance was maintained, as it represents the proportion of fallers and non-fallers commonly presented in the literature [38]. Given the encouraging findings of this study, we recommend applying this model to a more extensive database, even though the metrics and results of this study have confirmed the SOM was effectively able to recognize and cluster the presented patterns, even in a relatively modest but adequate sample size.

A conventional statistical method (*t*-test) was used to validate the SOM results, and again, no significant between-group difference was found (fallers and non-fallers). Visually, this result can be confirmed by the radar graphs, which show a marked similarity between the participant groups. This result corroborates the topographic and quantization errors found by SOM, confirming the absence of a significant difference between the two groups initially considered in this study based on the self-report of just one fall episode in the 12 months before data collection.

Corroborating these results, older adults with self-reports of only one fall in the last year did not exhibit significantly different patterns from those without self-reported falls when considering most kinematic variables under dual-task conditions [39]. These results are reinforced by another study in which it was demonstrated that older adults who self-reported only one fall in the last 12 months did not present any differences from non-faller older adults in terms of mean values for gait speed, cadence, step length, stride time, single-leg stance time, or variability of stride time during a dual-task gait [18].

Thus, upon examining clinical and functional characteristics in this study, it was found that these individuals categorized as fallers and non-fallers do not exhibit significant clinical and functional differences that warrant their classification in this manner. Considering a fall prevention point of view, these results may suggest that older adults with this type of functionality pattern could be approached similarly from an operational point of view, thus facilitating a collective approach process.

On the other hand, this study encourages a different approach to older adults who self-report two or more falls within 6 or 12 months and are considered recurrent fallers. It is

believed that these individuals may present more striking functional and clinical characteristics and could be efficiently identified by this unsupervised self-organization algorithm.

The secondary aim of this study was to determine which outcomes better discriminate community-dwelling older adults with and without self-reported falls by an unsupervised algorithm. The second premise of this study was confirmed, especially for variables that denote an increase in the cost of cognitive demands regarding a motor task (TUG) or in concurrent cognitive tasks (such as TMT A and B or Stroop test versions Black and White versus Colored). These outcomes proved to be capable of signaling a better grouping capacity of the participants in this study.

However, it is worth highlighting again that based on the sensitivity, precision, F1-score, and accuracy metrics, these groups are not necessarily fallers and non-fallers as expected. With this result, it is only possible to infer that these variables could divide the study participants into two groups with distinct motor-cognitive characteristics. It is essential to highlight that SOM is an unsupervised learning algorithm, i.e., an ML algorithm designed to initially deal with data sets in which samples do not have previously assigned labels or categories. Unlike supervised algorithms, in which the training data includes labeled input–output pairs, unsupervised algorithms explore the internal structure of the data to discover whether there are possible hidden patterns, clusters, or relationships without external guidance. The primary purpose of an unsupervised algorithm is to explore the intrinsic structure of the data and find useful information without the need for pre-defined labels or categories [13]. However, we chose to complement this study with metrics such as sensitivity, precision, F1-score, and accuracy that could present the performance of the SOM compared with the labels initially assigned to participants based on self-reported falls in the 12 months preceding the study.

On the other hand, the outcomes that denote functional performance related to postural balance did not show discriminatory capacity among the participants of this study, signaling that they may not be the best way of distinguishing older adults with and without self-reported falls. A systematic review on the use of the Functional Reach Test (FRT), Single-Leg Stance Test, and Tinetti Performance-Oriented Mobility Assessment (POMA) to predict falls in older adults showed that these clinical tests of postural balance should not replace a comprehensive assessment of the risk of falls and, therefore, should be incorporated into practice only to identify and track balance impairments in older adults [4].

In the same field of research, another study [7] employed SOM and k-means clustering (KM) to determine which gait characteristics best cluster individuals based on their age groups, which ranged from 10 to over 50 years old. The researchers demonstrated that these ML approaches efficiently clustered the main gait features based on the age of the participants. According to the authors, these findings can be helpful, especially for inexperienced professionals unlikely to identify subtle differences in walking metrics, aiming to recognize functional ability and physical performance [7]. Thus, these clustering methods can be valuable in supporting and guiding healthcare practitioners' evaluation and intervention processes based on age, history of falls, and functional or cognitive functioning.

It is worth noting, however, that the present study's results cannot be generalized to older adults with reports of recurrent falls or to dependent older adults due to the characteristics of the sample used. However, these input data allowed us to demonstrate the usefulness of the unsupervised method, such as the SOM, in recognizing nuances between the proposed variables and groups, even with homogeneous data, as shown in Table 1. It is also noteworthy that the results of this study are limited to outcomes from community-dwelling older adults assessed cross-sectionally; thus, we suggest employing this method to identify longitudinally the functional and cognitive patterns presented by older adults at risk of falls. Although the SOM algorithm is usually intended to work effectively with cross-sectional data [7,8], the valuable insights of this cross-sectional study encourage further longitudinal investigation, especially in older adults at risk of falls.

Age-associated changes in walking parameters are relevant to recognizing functional capacity and physical performance. However, the sensible nuances of slightly different

walking parameters are hardly noticeable by inexperienced observers, especially regarding the age-associated changes in gait relevant to recognizing functional capacity and physical performance. Due to the complexity of this type of evaluation, another study employing the SOM and k-means clustering (KM) showed this type of ML method efficiently clustered the principal gait characteristics according to age. The authors concluded that these clustering methods, when applied to the cross-sectional dataset, provide valuable information to healthcare professionals concerning the subject's physical performance related to age, supporting and guiding the physical evaluation.

Finally, the fundamental strength of this study is the application of the SOM algorithm as an intelligent self-organization method capable of mapping a set of functional and clinical data of high dimensions and complexity in an unsupervised manner in a two-dimensional topological space. Furthermore, this type of intelligence algorithm and the clinical and functional criteria used may distinguish between fallers and non-fallers among older adults.

## 5. Conclusions

The topological mappings generated by SOM revealed that outcomes such as functional mobility, postural balance, cognitive performance, self-efficacy concerning balance, concern about falls, and mood do not represent robust clinical and functional relationships capable of distinguishing community-dwelling older adults with and without self-report of a single fall in 12 months, as confirmed by conventional statistical methods. Nevertheless, the SOM results suggest that the cost of cognitive demands regarding a motor task (TUG) and the effect of the TMT-B regarding TMT-A could potentially identify functional and cognitive patterns in community-dwelling older adults. Therefore, in clinical practice, identifying patterns involving the interaction between cognition and motor skills, even in once-only faller older adults, can be an efficient approach to assessment and, consequently, to compound intervention programs to prevent falls in this population.

**Author Contributions:** Conceptualization, M.L.S.A. and F.T.-S.; methodology, M.L.S.A., R.S., R.R.C., F.B. and F.T.-S.; software, M.L.S.A., R.S., M.A.S., P.R.V.M. and F.T.-S.; validation, M.L.S.A., R.S., M.A.S., P.R.V.M., R.R.C., F.B. and F.T.-S.; formal analysis, M.L.S.A., R.S., R.R.C., M.A.S., P.R.V.M. and F.T.-S.; investigation, M.L.S.A. and F.T.-S.; resources, M.L.S.A., R.R.C. and F.T.-S.; data curation, F.T.-S.; writing—original draft preparation, M.L.S.A., A.O.C., R.S., M.A.S., P.R.V.M., A.A.M.S., R.R.C., F.B. and F.T.-S.; writing—review and editing, M.L.S.A., A.O.C., R.S., M.A.S., P.R.V.M., A.A.M.S., R.R.C., F.B. and F.T.-S.; visualization, M.L.S.A., A.O.C., R.S., M.A.S., P.R.V.M., A.A.M.S., R.R.C., F.B. and F.T.-S.; supervision, F.T.-S.; project administration, F.T.-S.; funding acquisition, F.T.-S. All authors have read and agreed to the published version of the manuscript.

**Funding:** The National Council for Scientific and Technological Development (CNPq) funded Trombini-Souza's research (Grant number: MCT/CNPq 423805/2016–9) and the Almeida's scientific initiation scholarship (Grant number: PIBIC 143082/2020-5). The *Coordenação de Aperfeiçoamento de Pessoal de Nível Superior—Brasil* (CAPES) funded Caldas's postdoctoral scholarship (Finance Code: 001). The APC was funded by the University of Pernambuco—UPE.

**Institutional Review Board Statement:** This study was conducted in accordance with the Declaration of Helsinki and approved by the Research Ethics Committee of the University of Pernambuco (CAAE: 71192017.0.0000.5207, Opinion Number: 2.415.658).

**Informed Consent Statement:** Written informed consent was obtained from all subjects involved in the study.

**Data Availability Statement:** The data used to support this study's findings are available from the corresponding authors upon request.

**Acknowledgments:** The authors are grateful to the National Council for Scientific and Technological Development (CNPq) for financing Trombini-Souza's research (Grant number: MCT/CNPq 423805/2016–9) and Almeida's scientific initiation scholarship (Grant number: PIBIC 143082/2020-5). The authors are also grateful to the *Coordenação de Aperfeiçoamento de Pessoal de Nível Superior—Brasil* (CAPES) for financing Caldas's postdoctoral scholarship (Finance Code: 001).

**Conflicts of Interest:** The authors declare no conflicts of interest. The funders had no role in the design of the study; in the collection, analyses, or interpretation of data; in the writing of the manuscript; or in the decision to publish the results.

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
