# Peer review of "Clinical and Functional Outcomes in Faller and Non-Faller Older Adults Clustered by Self-Organizing Maps: A Machine-Learning Approach"

_applsci, doi:10.3390/app14167093_

Round 1

Reviewer 1 Report

Comments and Suggestions for Authors

Thank you for the opportunity to review the authors' manuscript.

I have read the manuscript carefully.

And I found some concerns.

Details are provided below.

Falls prevention studies are usually longitudinal studies.

This is because it is important to know whether the results of a study can predict falls.

It is not desirable that this is not mentioned as a limitation of the studies.

The age range of 60 to 80 years is expected to have large differences in physical and cognitive function. In addition, there is no mention of gender differences or comorbidities. Please provide a comparison of baseline characteristics between the falls and non-falls groups. Otherwise, sampling bias cannot be ruled out.

I also appreciate the reference to sample size. However, it is unclear whether you are referring to the sample size required for the "t-test", "Mann-Whitney U test", and "SOM algorithm" described below. Please clarify.

Table 1. should include p-values and effect sizes.

Figure 1. has small text and is difficult to read. Please correct it.

The text explaining the contents of Table 2 talks about sensitivity, but Table 2 shows p-values. I believe there is a discrepancy in content.

In their conclusion, the authors state "SOM showed that the cost of cognitive demands on a motor task or other primary cognitive task could potentially identify functional patterns or However, given the number of subjects and the cross-sectional design of the study, I do not think this much can be said.

The terms "older adults" and "older people" are mixed up.

Please unify them.

That is all.

Author Response

We greatly appreciate the reviewer’s careful review and thoughtful comments. We have carefully revised the manuscript according to the reviewer’s insightful comments and provided a point-by-point response letter. All the reviewer’s comments are answered and complemented in the revised manuscript, highlighted in yellow. Some of the comments will be discussed here. Below, we respond to your remarks on a point-by-point basis.

Comment 1. Falls prevention studies are usually longitudinal studies. This is because it is important to know whether the results of a study can predict falls. It is not desirable that this is not mentioned as a limitation of the studies.

Response: We agree with the reviewer and appreciate the suggestion. We have added the following part in the paragraph regarding the study’s limitation: “It is also noteworthy that the results of this study are limited to outcomes from communi-ty-dwelling older adults assessed cross-sectionally; thus, we suggest employing this method to identify longitudinally the functional and cognitive patterns presented by older adults at risk of falls. Although the SOM algorithm is usually intended to work with cross-sectional data effectively [7,16], the valuable insights of this cross-sectional study encourage further longitudinal investigation, especially in older adults at risk of falls. (Lines 385-391)

Comment 2. The age range of 60 to 80 years is expected to have large differences in physical and cognitive function. In addition, there is no mention of gender differences or comorbidities. Please provide a comparison of baseline characteristics between the falls and non-falls groups. Otherwise, sampling bias cannot be ruled out.

Response: We agree with this reviewer that older adults aged 60 to 80 are generally expected to have significant differences in physical and cognitive function. However, some methodological precautions were taken into account in this study to avoid any potential functional and cognitive discrepancy between participants. Firstly, it was required that all participants were community-dwelling older adults. Furthermore, from the functional aspect, all participants must be able to walk uninterruptedly for a distance of 60 m at a self-selected speed of at least 1 m/s (without the help of third parties or walking aids) and achieve a score ≥ 52 points (up to a maximum of 56 points) on the Berg Balance Scale. Regarding cognitive function, all participants must have a score of ≥ 24 points on the Mini-Mental State Exam (MMSE) (score maximum of 30 points) for those with > 4 years of formal education or ≥ 18 points for those with < 4 years of formal education. To add even more robustness to the participant recruitment stage and to homogenize the sample as much as possible, potential participants were excluded if they: (i) had any chronic health condition for which physical exercise was contraindicated, (ii) had any upper or lower limb fracture in the last six months, (iii) had evidence of any surgical procedure on the knees, ankles, and hips or muscle injuries in the last six months, or (iv) had a diagnosis of uncontrolled diabetes. The “Participants” subsection presents all these methodological aspects as eligibility criteria. To confirm that all participants did not present significant differences between them in functional and cognitive aspects, despite the different stage ranges, we have divided the sample and performed Kruskal-Wallis tests considering five age groups as follows: 60-64 y.o. (n = 19; 31,7%); 65-69 y.o. (n = 21; 35,0%); 70-74 y.o. (n = 14; 23,3%); 75-80 y.o. (n = 6; 10,0%). The results for functional and cognitive outcomes were: gait speed (p = 0.446); conventional TUG (p = 0.455); cognitive TUG (p = 0.621); dual-task effect (TUG) (p = 0.398); sit to stand from a chair for 5 times (p = 0.245); anterior functional reach test (p = 0.282); ABC scale (p = 0.718) and FES (p = 0.133). This part was inserted in the result section as follows: “Firstly, participants were divided into four age groups to verify any significant difference in physical and cognitive function inherent to age, as follows: 60-64 years old (y.o.) (n = 19; 31,7%); 65-69 (n = 21; 35,0%); 70-74 y.o. (n = 14; 23,3%); 75-80 y.o. (n = 6; 10,0%). The Kruskal-Wallis test for independent groups confirmed no significant functional and cognitive differences as follows: gait speed (p = 0.446); conventional TUG (p = 0.455); cognitive TUG (p = 0.621); dual-task effect (TUG) (p = 0.398); sit to stand from a chair for 5 times (p = 0.245); anterior functional reach test (p = 0.282); ABC scale (p = 0.718) and FES (p = 0.133).” (Lines 194-200)

Comment 3. I also appreciate the reference to sample size. However, it is unclear whether you are referring to the sample size required for the “t-test”, “Mann-Whitney U test”, and “SOM algorithm” described below. Please clarify.

Response: We apologize to the reviewer for the confusion regarding the definition of the sample size for the original study (EQUIDOSO-I) and the present one. To mitigate this misunderstanding, we think it is best to include a description of the sample size calculation of the EQUIDOSO-I study in the “Participants” section, as follows: “For the sample size calculation in the EQUIDOSO-I study [18], the gait speed under du-al-task [22] (the primary outcome) was adopted. Aiming to achieve a minimum clinically important difference of 0.05 m/s, an effect size of 0.20 [24], a test power of 95% (1 - β), an alpha of 0.05, and a repeated-measures F-statistic design, with between- and with-in-subject and interaction effects were considered. The initial sample size of EQUIDOSO-I was 48 individuals, which was increased by 20% to account for potential sample loss. Thus, a total of 60 participants was assessed to adequately provide the required power for traditional statistical tests employed in the EQUIDOSO-I study. The sample size was calculated using the G* Power 3 program [25].” (Lines 117-125) As requested, the references to sample size calculation are cited in the article and at the end of this comment. Response on the SOM study’s sample size (secondary analysis from EQUIDOSO-I dataset):

For this response, it is essential to highlight that the primary goal of this study was to examine how the intelligent SOM algorithm clusters older adult populations with or without fall history based on data already collected (data secondary analysis). According to Kianga, Hub, and Fisher (2007)4, unlike conventional statistics, neural networks, such as SOM, are not dependent on statistical assumptions. Also, neural networks are inherently less dependent on sample size if the sample is reasonably representative of the underlying population. Despite the SOM’s robustness to sample size, ensuring that the sample size sufficiently represents the target population is essential. Thus, the sample from EQUIDOSO-I provides a diverse and representative dataset of older adults aged 60 to 80, with or without fall history, which is crucial for clustering analysis. Furthermore, the primary focus in SOM analysis is the convergence of the model rather than adherence to traditional statistical power considerations. Our results indicate that the SOM successfully converged with the given sample, as evidenced by the stability and interpretability of the clusters formed. The convergence of the SOM models was achieved using the proposed dataset, demonstrating that the model’s results are consistent and reliable. The large number of variables in the dataset further provided this convergence, as high-dimensional data can help the SOM better capture and cluster the inherent patterns. Additionally, the final metrics of the SOM model, including cluster quality and internal validation measures, support the robustness of the results, including the assessing metrics such as the Quantization and Topographic Errors, which indicate the clustering quality. The consistency of the results across multiple runs reinforces the validity of the findings. However, these data regarding the test of several intelligent models we have carried out are in consideration as an article submitted to another journal in artificial intelligence. In summary, while traditional sample size calculations are crucial for conventional statistical hypothesis testing, SOM’s performance is influenced by the representativeness of the data and the model’s convergence. Thus, the sample size used in this study was sufficient to achieve meaningful clustering, and the robustness of the results is evident from the final metrics. Therefore, we opted not to go into too much detail on the sample calculation section topic to stay true to the study’s goal. We recognize that methodological articles analyzing various model types and intelligent algorithms are better suited for discussing this subject. However, we included the following sentences in the sample calculation section to help readers understand the basic and essential methodological considerations regarding the SOM data required. “Thus, in this secondary analysis, using the SOM, we employed a high-dimensional and representative dataset of 60 community-dwelling older adults aged 60 to 80, with or without fall history, to guarantee the quality of the final clustering model. This methodological feature was sufficient to ensure that the SOM attained sufficient convergence, improved data-innate patterns, improved participant grouping, and stability and interpretability of the generated clusters.” (Lines 136-141)

[18] Trombini-Souza, F.; de Maio Nascimento, M.; da Silva, T.F.A.; de Araujo, R.C.; Perracini, M.R.; Sacco, I.C.N. Dual-task training with progression from variable- to fixed-priority instructions versus dual-task training with variable-priority on gait speed in community-dwelling older adults: A protocol for a randomized controlled trial : Variable- and fixed-priority dual-task for older adults. BMC Geriatr 2020, 20, 76, doi:10.1186/s12877-020-1479-2.

[22] Silsupadol, P.; Shumway-Cook, A.; Lugade, V.; van Donkelaar, P.; Chou, L.S.; Mayr, U.; Woollacott, M.H. Effects of single-task versus dual-task training on balance performance in older adults: a double-blind, randomized controlled trial. Archives of physical medicine and rehabilitation 2009, 90, 381-387, doi:10.1016/j.apmr.2008.09.559.

[25] Perera, S.; Mody, S.H.; Woodman, R.C.; Studenski, S.A. Meaningful change and responsiveness in common physical performance measures in older adults. J Am Geriatr Soc 2006, 54, 743-749, doi:10.1111/j.1532-5415.2006.00701.x.

Kiang, M.Y.; Hu, M.Y.; Fisher, D.M. The effect of sample size on the extended self-organizing map network—A market segmentation application. Computational Statistics & Data Analysis. Volume 51, Issue 12, 15 August 2007, Pages 5940-5948.

Comment 4. Table 1 should include p-values and effect sizes.

Response: As suggested by the reviewer, p-values and effect sizes were included in Table 1.

 Comment 5. Figure 1 has small text and is difficult to read. Please correct it.

Response: We agree with this reviewer. Thus, the font size of the Figure 1 was increased.

Comment 6. The text explaining the contents of Table 2 talks about sensitivity, but Table 2 shows p-values. I believe there is a discrepancy in content.

Response: We apologize for the confusion in naming the last column in Table 2. “Value” does not refer to “p-value” as it seemed. In fact, the last column of this table contains the “Metric value”. Thus, this column was renamed as “Metric value”.

Comment 7. In their conclusion, the authors state “SOM showed that the cost of cognitive demands on a motor task or other primary cognitive task could potentially identify functional patterns”. However, given the number of subjects and the cross-sectional design of the study, I do not think this much can be said.

Response: We thank the reviewer for your thoughtful comments on this topic. We appreciate your concern regarding interpreting our results, given the study’s cross-sectional design and sample size. To soften this conclusion, we used the word “suggest” and the expression “could potentially”, as follows: “Nevertheless, SOM results suggest that the cost of cognitive demands regarding a motor task (TUG) and the effect of the TMT-B regarding the TMT-A could potentially identify functional patterns or relationships in community-dwelling older adults.(Lines 413- 416)

We further included the following statements in the study’s limitations: “It is also noteworthy that the results of this study are limited to outcomes from community-dwelling older adults assessed cross-sectionally; thus, we suggest employing this method to identify longitudinally the functional and cognitive patterns presented by older adults at risk of falls. Although the SOM algorithm is usually intended to work with cross-sectional data effectively [7,16], the valuable insights of this cross-sectional study encourage further longitudinal investigation, especially in older adults at risk of falls.(Lines 385-391)

Furthermore, we would like to clarify that the main functionality of SOM is its capability to operate on cross-sectional datasets and, consequently, identify and cluster patterns within such data. We highlighted the primary objective of our study was to utilize SOM to cluster latent functional, cognitive, and clinical patterns in the baseline dataset of a longitudinal study before the intervention to improve functional, clinical, and cognitive outcomes in older adults with or without a history of falls. Then, from this cross-sectional dataset, the cost of cognitive demands regarding a motor task (TUG) and the effect of the TMT-B regarding the TMT-A were the outcomes with better ability to evidence a pattern between the two groups. It is also noteworthy that the SOM algorithm is intended to work with cross-sectional data, where it effectively clusters and identifies patterns without the need for longitudinal data. The metrics generated by SOM in this study, including cluster quality measures and internal validation metrics, demonstrated robust and consistent results. These metrics indicate that the identified patterns are meaningful and provide insights into the relationship between cognitive demands and motor performance. Although our sample size appears relatively modest, it was sufficient for the SOM to identify and cluster patterns effectively. The consistency of the results across different metrics and models supports the validity of our findings. A similar study, recently published in a reputed journal in this area (IEEE Access), employing this type of machine learning of clustering with comparable size, features, and design (cross-sectional) has successfully identified meaningful patterns, reinforcing the adequacy of our sample size for the analysis (Sarai et al., 2022). Thus, our conclusions are based on the observed clustering patterns and the robustness of the SOM metrics. Although the study design is cross-sectional, the patterns identified by the SOM can provide valuable insights and inform future research directions. The results suggest potential functional patterns that merit further investigation. In conclusion, we believe that using SOM in this cross-sectional study has provided valuable insights into the relationship between cognitive demands and motor tasks. The robust metrics generated by the SOM support the validity of our findings and the potential implications outlined in our conclusions. Once again, we appreciate your consideration and feedback and hope this clarification addresses your concerns.

Sarai, R.; Trombini-Souza, F.; Moura, V.T.G.; Caldas, R.; Buarque, F. Classification of Older Adults Undergoing Two Dual-Task Training Protocols Based on Artificial Intelligent Methods. IEEE Access 2022, 10, 3066-3073, doi:10.1109/ACCESS.2021.3139527.

Comment 8. The terms “older adults” and “older people” are mixed up. Please unify them.

Response: We thank the reviewer for pointing out this need for standardization. As suggested, “older people” was changed to “older adults” throughout the text.

Reviewer 2 Report

Comments and Suggestions for Authors

This research aimed to identify patterns in older adults who fall compared to those who don't. Using a an unsupervised machine learing algorithm (SOM), the authors analyzed clinical, functional, and cognitive data from 60 older adults.

The study appears to be well-written, but it could be enhanced by addressing the following points:

  • Incorporate a more comprehensive review of relevant studies in both the introduction and discussion sections to provide a stronger context for the research and to highlight the study's contribution to the field, adding a related works section.
  • Enhance readability by adding subtitles (i.e., database, networks and so on…) within the materials and methods section to clearly delineate different aspects of the methodology.
  • Please provide a better quality of Figure 3
  • Given the potential impact of class imbalance on the study's findings, did the authors consider employing data augmentation techniques to mitigate this issue?

Author Response

We greatly appreciate the reviewer’s careful review and thoughtful comments. We have carefully revised the manuscript according to the reviewer’s insightful comments and provided a point-by-point response letter. All the reviewer’s comments are answered and complemented in the revised manuscript, highlighted in yellow. Some of the comments will be discussed here. Below, we respond to your remarks on a point-by-point basis.

This research aimed to identify patterns in older adults who fall compared to those who don’t. Using an unsupervised machine learning algorithm (SOM), the authors analyzed clinical, functional, and cognitive data from 60 older adults. The study appears to be well-written, but it could be enhanced by addressing the following points:

Comment 1. Incorporate a more comprehensive review of relevant studies in both the introduction and discussion sections to provide a stronger context for the research and to highlight the study’s contribution to the field, adding a related works section.

Response: We agree with this reviewer that both sections may be improved regarding this aspect. As suggested, we have provided some studies on this topic in the introduction section as follows:  “Machine learning (ML) is at the core of the Big Data Revolution because of its ability to learn from data and provide data-driven insights, decisions, and predictions. It promises to transform how we live, work, and think by optimizing processes, empowering insight discovery, and improving decision-making [9]. In this context, several ML-based models, such as random forest (RF), logistic regression (LR), support vector machine (SVM), light gradient boosting machine (LightGBM), and extra trees (ET) have been applied to analyze functional dependency in basic activities of daily living (ADL) and instrumental activities of daily living (IADL) in middle-aged and older adults [10]. ML-based models have also been used to analyze motion, clinical, cognitive, and functional outcomes [11,12]. Ensemble techniques, such as AdaBoost, clustering models (e.g., k-means), and SVM, have presented promising measurement properties in identifying and differentiating the functional features in community-dwelling older adults undergoing exercise protocols [12].” (Lines 58-70) In the discussion section, we have added the following part: “In the same field of research, another study [7] employed SOM and k-means clustering (KM) to determine which gait characteristics best cluster individuals based on their age groups, which ranged from 10 to over 50 years old. The researchers demonstrated that these ML approaches efficiently clustered the main gait features based on the age of the participants. According to the authors, these findings can be helpful, especially for inexperienced professionals unlikely to identify the subtle differences in walking metrics aiming to recognize functional ability and physical performance [7]. Thus, these clustering methods can be valuable in supporting and guiding healthcare practitioners’ evaluation and intervention processes based on age, history of falls, and functional or cognitive functioning.” (Lines 371-380)

Comment 2. Enhance readability by adding subtitles (i.e., database, networks and so on…) within the materials and methods section to clearly delineate different aspects of the methodology.

Response: We appreciate the suggestion. As suggested, we have added the following subheadings: Study design and ethical aspects; Participants; Database; Analysis. (Lines 86, 98, 127, and 149)

Comment 3. Please provide a better quality of Figure 3

Response: We agree with the reviewer. Image quality has been improved and increased to 300 dpi.

Comment 4. Given the potential impact of class imbalance on the study’s findings, did the authors consider employing data augmentation techniques to mitigate this issue?

Response: We thank the reviewer for bringing this question up for consideration. Throughout the initial analyses, we also questioned this result. We believed that an accuracy of approximately 50% may have been the imbalance between the number of participants in the two groups, as the group of fallers consisted of 22 participants while the non-fallers group contained 38 older adults. However, when analyzing the results of models with balanced samples (22 fallers and 22 non-fallers), no significant improvement in the magnitude of the model accuracy was observed. Furthermore, the F1-score value was not significantly modified with or without the imbalanced sample. F1-score is particularly useful given a sample size imbalance in classification problems, as it considers both precision and recall, offering a more balanced view of model performance regardless of class imbalance. Therefore, the model with a sample imbalance was maintained, as it represents the proportion of fallers and non-fallers commonly presented in the literature. Although this imbalance appears to interfere with accuracy, the proportion of approximately 30% is representative of the prevalence of older adult fallers in the general population. As suggested by this reviewer, we rephrased this part of the discussion: “Furthermore, the F1-score value was not significantly modified with or without the imbalanced sample. It is worth highlighting that the F1-score (the harmonic mean of precision and sensitivity) is a valuable metric in classification problems with sample size imbalances as it considers both precision and recall, providing a more balanced view of model performance regardless of class imbalance. Therefore, the model with a sample imbalance was maintained, as it represents the proportion of fallers and non-fallers commonly presented in the literature [39]. Given the encouraging findings of this study, we recommend applying this model to a more extensive database, even though the metrics and results of this study have confirmed the SOM was effectively able to recognize and cluster the presented patterns, even in a relatively modest but adequate sample size.(Lines 304-314)

Reviewer 3 Report

Comments and Suggestions for Authors

Hi. This is an important topic. 

1. I do not understand the last sentence of the abstract: "The increased cost 28 of cognitive demands regarding a motor task or another primary cognitive task could discriminate 29 the functional patterns in community-dwelling older adults.". I guess more will not understand. please revise

2. Machine learning and/or AI should be part of the article title. Otherwise, some potential readers will skip. the term "self-organizing maps" is not familiar enough

3. Figure 03 recommended as supplementary material. Once again... most clinicians reading do not understand / are truly interested in the "machinery" described but rather in the clinical conclusions

4. I think that the conclusions should not only repeat what is already written but try to go one step further to clinical implications

Author Response

We greatly appreciate the reviewer’s careful review and thoughtful comments. We have carefully revised the manuscript according to the reviewer’s insightful comments and provided a point-by-point response letter. All the reviewer’s comments are answered and complemented in the revised manuscript, highlighted in yellow. Some of the comments will be discussed here. Below, we respond to your remarks on a point-by-point basis.

Comment 1. I do not understand the last sentence of the abstract: “The increased cost of cognitive demands regarding a motor task or another primary cognitive task could discriminate the functional patterns in community-dwelling older adults.”. I guess more will not understand. Please revise.

Response: We thank the reviewer for pointing out this aspect. The sentence was really confusing. We rewrite it as follows: “The increased cost of cognitive demands regarding a motor task (Timed Up and Go Test) and the effect of the Trail Making Test part B regarding A could discriminate the functional patterns in community-dwelling older adults.” (Lines 30-32). We have also improved the way this point is written in the discussion as follows: “The second premise of this study was confirmed, especially for variables that denote an increase in the cost of cognitive demands regarding a motor task (TUG) or in concurrent cognitive tasks (such as TMT A and B or Stroop test versions black and white versus colored)”. (Lines 342-345).

Comment 2. Machine learning and/or AI should be part of the article title. Otherwise, some potential readers will skip the term “self-organizing maps” is not familiar enough.

Response: We agree, and we thank the reviewer for the suggestion. The new title is as follows: “Clinical and functional outcomes in faller and non-faller older adults clustered by self-organizing maps: A machine-learning approach” (Lines 2-4).

Comment 3. Figure 03 recommended as supplementary material. Once again... most clinicians reading do not understand / are truly interested in the “machinery” described but rather in the clinical conclusions.

Response: We agree with the reviewer that most clinicians reading do not understand or are genuinely interested in the “machinery” described but rather in the clinical conclusions. Therefore, in this new version of the manuscript, we better explain how to interpret this type of graph since the other reviewers asked to improve the quality of this figure presented in the body of the article. “The more diametrically opposed the darker-colored neurons are, the greater the discrimi-native potential of a given outcome. In the topographic distribution concerning the cogni-tive TUG (TUGcog_T1), it is possible to see that the neurons representing the individuals with a pattern of the longer execution time of this dual-task test are concentrated in the lower left corner (dark yellow), while the participants who performed this test in a shorter time (better performance) are located in the upper right corner (dark blue), i.e., in opposite directions. In the topographic distribution of the effect of the cognitive TUG compared to the conventional TUG (DualTaskEffect_T1), the neurons representing the individuals with the highest cost pattern are concentrated in the lower left corner (dark blue), while those with the lowest cost are located in the upper right corner (green and yellow). Finally, the topographic distribution of neurons referring to the effect of the TMT-B regarding the TMT-A (TMT_effect) shows that individuals with less cognitive difficulty (dark yellow) are concentrated in the upper left corner, while participants with greater cognitive difficulty are located in the bottom right corner (dark blue).” (Lines 255-270). After including this explanation in the article body, we respectfully decided to leave this rich and innovative figure in the revised version of the manuscript so that other readers could also benefit from this type of graph knowledge. Many other readers choose to view this kind of graph straight in the main article rather than searching the supplemental material for more information. Once again, we appreciate your consideration and feedback and hope this clarification regarding this image addresses your concerns.

Comment 4. I think that the conclusions should not only repeat what is already written but try to go one step further to clinical implications.

Response: We agree entirely with the reviewer. Therefore, we complement the conclusion of this study as follows: “Therefore, in clinical practice, identifying patterns involving the interaction between cognition and motor skills, even in once-only faller older adults, can be an efficient approach to assessment and, consequently, to compound intervention programs to prevent falls in this population.” (Lines 416-420)

Round 2

Reviewer 1 Report

Comments and Suggestions for Authors

I thank the authors for their prompt corrections and thoughtful responses.

I have verified that the appropriate corrections have been made.

Author Response

We are pleased that the adjustments fulfilled all of the reviewer’s expectations. Thanks again for the careful review and thoughtful comments.

Reviewer 2 Report

Comments and Suggestions for Authors

The authors have made all the requested revisions; therefore, I recommend the article for publication.

Author Response

(The authors gave the same response as above.)
